# Decay times of atmospheric acoustic-gravity waves after deactivation of wave forcing

Nikolai M. Gavrilov[1], Sergey P. Kshevetskii[1,2], Andrey V. Koval[1,3]

[1]Atmospheric Physics Department, Saint-Petersburg State University, Saint Petersburg, 199034, Russia
[2]Physics Department, Immanuel Kant Baltic Federal University, Kaliningrad, 236016, Russia
[3]Meteorological Forecast Department, Russian State Hydro-meteorological University, Saint Petersburg, 192007, Russia

*Correspondence to*: Nikolai M. Gavrilov (n.gavrilov@spbu.ru)

**Abstract.** High-resolution numerical simulations of non-stationary nonlinear acoustic-gravity waves (AGWs) propagating
upwards from surface wave sources are performed for different temporal intervals relative to activation/deactivation times of
the wave forcing. After activating surface wave sources, amplitudes of AGW spectral components reach a quasi-stationary
state. Then the surface wave forcing is deactivated in the numerical model, and amplitudes of vertically traveling AGW
modes quickly decrease at all altitudes due to discontinuations of the upward propagation of wave energy from the wave
sources. However, later the standard deviation of residual and secondary wave perturbations experiences slower quasi-
exponential decrease. High-resolution simulations allowed, for the first time, estimating the decay times of this wave noise
produced by slow residual, quasi-standing and secondary AGW spectral components, which vary between 20 and 100 hrs
depending on altitude and the rate of wave source activation/deactivation. The standard deviations of the wave noise are
larger for the case of sharp activation/deactivation of the wave forcing compared to the gradual processes. These results
show that transient wave sources may create long-lived wave perturbations, which can form a background level of wave
noise in the atmosphere. This should be taken into account in parameterizations of atmospheric AGW impacts.

## 1 Introduction

Recently, acoustic-gravity waves (AGWs) are believed to exist almost permanently in the atmosphere (Siefring et al., 2010;
Snively et al., 2013; Wei et al., 2015; Lay, 2018; Meng et al., 2019). Observations detect regular AGW presence up to high
atmospheric altitudes (e.g., Djuth et al., 2004; Park et al., 2014; Trinh et al., 2018). Modeling of general circulation
demonstrated AGWs capabilities of transferring energy and momentum from tropospheric wave sources to higher
atmospheric levels (e.g., Medvedev and Yiğit, 2019). Non-hydrostatic models of the general circulation of the atmosphere
revealed that AGWs are permanently existent at all atmospheric heights (e.g. Yiğit et al., 2012b).

Many AGWs detected in the atmosphere are excited in the troposphere (Fritts and Alexander, 2003; Snively, 2013; Yiğit
et al., 2014). AGWs can be produced by interactions of winds with mountains (e.g., Gossard and Hooke, 1975), atmospheric
jet streams and fronts (e.g., Gavrilov and Fukao, 1999; Dalin et al., 2016), thunderstorms and cumulus clouds (Siefring et al.,

2010; Blanc et al., 2014; Lay, 2018), convective regions and shear flows (Townsend, 1966; Fritts and Alexander , 2003; Vadas and Fritts, 2006), typhoons (Wu et al.,2015), volcanoes (De Angelis et al., 2011), waves on the sea surface (Godin et al., 2015), by explosions at the Earth's surface (Meng, 2019), earthquakes (Rapoport et al., 2004), tsunamis (Wei at al., 2015), different objects moving in the atmosphere  (Afraimovich et al., 2002), big fires, etc. Some AGWs can be generated by mesoscale turbulence in the atmosphere (Townsend, 1965; Medvedev and Gavrilov, 1995). These AGW sources are located mainly at tropospheric heights (Gavrilov and Fukao, 1999; Dalin, 2016).

Most wave sources listed above are non-stationary. They can be activated during initial time intervals, operate for some time, and then can be deactivated during final time intervals. The initial and final time intervals could be shorter or longer depending on the physical properties of particular wave sources. Non-stationary activating and deactivating wave sources can generate transient AGW pulses propagating upwards from the lower atmosphere, which require their analysis.

High-resolution numerical models are frequently used for studies of meso- and microscale processes in the atmosphere. For example, the Weather Research and Forecasting Non-hydrostatic Mesoscale Model (WRF, 2019) known also as the North American Mesoscale model, as well as the Regional Atmospheric Modeling System (RAMS) described by Pielke et al. (1992) and other similar models. Direct Numerical Simulation (DNS) and Large Eddy Simulation (LES) models (e.g., Mellado, 2018) should be mentioned in this context. Fritts et al. (2009, 2011) used a numerical model of Kelvin-Helmholtz instabilities, AGW breaking and generation of turbulence in atmospheric regions with fixed horizontal and vertical extents. They utilized a Galerkin-type algorithm for turning partial differential equations into equations for spectral series coefficients. Liu et al. (2009) simulated propagation of atmospheric AGWs and creation of Kelvin-Helmholtz billows. Yu et al. (2017) used a numerical model for AGWs propagating in the atmosphere from tsunamis.

Gavrilov and Kshevetskii (2013) studied nonlinear AGWs with a numerical two-dimensional model, which involved fundamental conservation laws. This model permitted non-smooth solutions of the nonlinear wave equations and gave the required stability of the numerical model (Kshevetskii and Gavrilov, 2005). A respective three-dimensional algorithm was introduced by Gavrilov and Kshevetskii (2014) to simulate nonlinear atmospheric AGWs. Gavrilov and Kshevetskii (2013. 2014) showed that after triggering wave forcing at the lower boundary of the numerical model, initial AGW pulses could reach high atmospheric levels in a few minutes. AGW phase surfaces are quasi-vertical initially, but later they become inclined to the horizon. AGW vertical wavelengths decrease in time and are close to their theoretical predictions after intervals of a few periods of wave forcing.

In this study, using the high-resolution nonlinear wave model developed by Gavrilov and Kshevetskii (2014), we continue simulating transient waves generated by non-stationary AGW sources at the lower boundary and propagating upwards to the atmosphere. The focus is AGW behavior after deactivations of wave sources in the model. After activating the surface wave source and disappearing initial wave pulses, AGW amplitudes tend to stabilize at all atmospheric altitudes. In this quasi-stationary state, the surface wave forcing is deactivated in the numerical model. After that, amplitudes of traveling AGW modes quickly decrease at all altitudes due to discontinuation of the upward propagation of wave energy

from the surface sources. We found, however, that after some time, the standard deviation of residual and secondary wave
perturbations experiences more slow exponential decrease with substantial decay times.

These results show that residual and secondary AGW modes produced by transient wave sources can exist for long time in the stratosphere and mesosphere and form a background level of wave noise there. AGW decay times and their dependences on parameters of the surface wave forcing are estimated for the first time.

## 2 Numerical model

In this study, we employed the high-resolution three-dimentional numerical model of nonlinear AGWs in the atmosphere developed by Gavrilov and Kshevetskii (2014). Currently, this model (called as AtmoSym) is available for free online usage (AtmoSym, 2017). The AtmoSym model utilizes the plain geometry and primitive hydrodynamic three-dimensional equations (Gavrilov and Kshevetskii, 2014):

$$\frac{\partial \rho}{\partial t} + \frac{\partial \rho v_\beta}{\partial x_\beta} = 0, \ \ \rho c_p \frac{dT}{dt} = \frac{dp}{dt} + \rho \varepsilon, \ \ p = \rho RT,$$

$$\frac{\partial \rho v_i}{\partial t} + \frac{\partial \rho v_i v_\beta}{\partial x_\beta} = -\frac{\partial p}{\partial x_i} - \rho g \delta_{i3} + \frac{\partial \sigma_{i\beta}}{\partial x_\beta}, \ \ i, \beta = 1, 2, 3 \, , \tag{1}$$

where $t$ is time; $p$, $\rho$, $T$ are pressure, density and temperature, respectively; $v_\beta$ are velocity components along the coordinate axes $x_\beta$; $\sigma_{i\beta}$ is the viscous stress tensor; $g$ is the acceleration due to gravity; $c_p$ is the specific heat capacity at constant pressure; $R$ is the atmospheric gas constant; $\varepsilon$ is the specific heating rate; $d/dt = \partial/\partial t + v_\beta \partial/\partial x_\beta$; repeating Greek indexes assume summation. Quantities $\sigma_{i\beta}$ and $\varepsilon$ in Eq. (1) contain stresses and heating rates produced by molecular viscosity and
heat conductivity (see details in Gavrilov and Kshevetskii, 2014). After numerical integration of Eq. (1), dynamical deviations (marked with primes below) from stationary background values $p_0$, $\rho_0$, $T_0$ and $v_{i0}$ are calculated:

$$p' = p - p_0; \ \ \rho' = \rho - \rho_0; \ \ T' = T - T_0; \ \ v'_i = v_i - v_{i0}. \tag{2}$$

The AtmoSym model takes into account dissipative and nonlinear processes that accompany AGW propagation. The model is capable to simulate such complicated processes as AGW instability, breaking and turbulence generation. Dynamical
deviations as defined in Eq. (2) describe both wave perturbations and modifications of background fields due to momentum and energy exchange between dissipating AGWs and the atmosphere. The background temperature $T_0(z)$ is obtained from the semi-empirical NRLMSISE-00 atmospheric model (Picone et al., 2002). Background dynamic molecular viscosity, $\mu_0$, and heat conductivity, $\kappa_0$, are estimated using the Sutherland's formulae (Kikoin, 1976):

$$\mu_0 = \frac{1.46 \times 10^{-6} \sqrt{T_0}}{1 + 110/T_0} \ \ \left( \frac{kg}{m \cdot s} \right) \tag{3}$$

$$\kappa_0 = \frac{\mu_0}{\text{Pr}}; \ \ \ \text{Pr} = \frac{4\gamma}{9\gamma - 5},$$

where $\gamma$ is the heat capacity ratio, *Pr* is the Prandtl number. The AtmoSym model involves also the mean turbulent thermal conductivity and viscosity having maxima of about 10 m$^2$s$^{-1}$ in the boundary layer and in the lower thermosphere, and a broad minimum of up to 0.1 m$^2$s$^{-1}$ in the stratosphere (Gavrilov and Kshevetskii, 2014). The upper boundary conditions at $z = h$ have the following form (Kurdyaeva et al., 2018):

$$\left(\frac{\partial T}{\partial z}\right)_{z=h} = 0, \quad \left(\frac{\partial v_1}{\partial z}\right)_{z=h} = 0, \quad \left(\frac{\partial v_2}{\partial z}\right)_{z=h} = 0, \quad \left(w\right)_{z=h} = 0, \tag{4}$$

where indices 1 and 2 correspond to horizontal directions, $w = v_3$ is vertical velocity. Conditions of Eq. (4) may cause reflections of AGWs coming from below. The upper boundary at the present study is set at $h = 600$ km, where molecular viscosity and heat conductivity are very high and reflected waves are strongly dissipated. Sensitivity tests reveal that the impact of conditions at the upper boundary as defined in Eq. (4) is negligible at altitudes $z < h - 2H$, where $H$ is the atmospheric scale height. Therefore, at altitudes of the middle atmosphere analyzed in this paper, the influence of the upper

boundary conditions (Eq. (4)) could be negligible. The lower boundary conditions at the Earth's surface have the following form (see Kurdyaeva et al., 2018):

$$\left(T'\right)_{z=0} = 0, \quad \left(v_1\right)_{z=0} = 0, \quad \left(v_2\right)_{z=0} = 0, \quad \left(w\right)_{z=0} = W_0 \cos(\sigma t - \vec{k} \cdot \vec{r}), \tag{5}$$

where $W_0$ and $\sigma$ are the amplitude and frequency of wave excitation, $\vec{k} = (k_1, k_2)$ is the horizontal wave vector, $\vec{r} = (x_1, x_2)$ is the position vector in the horizontal plane, $k_1$ and $k_2$ are the wavenumbers along horizontal axes $x_1$ and $x_2$,

respectively. The last relation for the surface vertical velocity in Eq. (5) serves as the source of plane AGW modes in the AtmoSym model. Such plane modes can represent spectral components of tropospheric dynamical processes. Their effects can be approximated by appropriate sets of effective spectral components of vertical velocity at the lower boundary (Townsend, 1965, 1966). Along horizontal axes $x_1$ and $x_2$, one can assume periodicity of wave fields

$$F(x_1, x_2, z, t) = F(x_1 + L_1, x_2 + L_2, z, t), \tag{6}$$

where $F$ denotes any of simulated hydrodynamic quantities, $L_1 = n_1\lambda_1$ and $L_2 = n_2\lambda_2$ are horizontal dimensions of the analyzed atmospheric region; $\lambda_1 = 2\pi/k_1$ и $\lambda_2 = 2\pi/k_2$ are wavelengths along axes $x_1$ and $x_2$, respectively; $n_1$ and $n_2$ are integers.

  In our simulations, the wave excitation in Eq. (5) is activated at the moment $t = t_a$ and then its amplitude $W_0$ does not change for some time. One should expect that at small amplitudes of wave source in Eq. (5), the numerical solutions in the

lower and middle atmosphere should tend at $t \gg t_a$ to a steady-state plane AGW modes corresponding to the traditional linear theory (e.g., Gossard and Hooke, 1975). Gavrilov et al. (2015) showed good agreement of ratios of simulated amplitudes of different wave fields with polarization relations of linear AGW theory (Gossard and Hooke, 1975) at $t \gg t_a$ at altitudes up to 100 km.

The novelty of the present study is deactivating the wave source in Eq. (5) at some moment $t = t_d$ after reaching the
described above quasi-steady solution. Previous simulations with the AtmoSym model showed that sharp activating the
surface wave source (Eq. (5)) could create an initial AGW pulse, which can reach high altitudes in a few minutes. To control
the rate of the wave source activation/deactivation, in the present simulations, we multiply the surface vertical velocity in Eq.
(5) by a function

$$q(t) = \begin{cases} \exp[-(t-t_a)^2 / s_a^2] & at & t \leq t_a \\ 1 & at & t_a < t < t_d \\ \exp[-(t-t_d)^2 / s_d^2] & at & t \geq t_d \end{cases} \tag{7}$$

where $s_a$ and $s_d$ are constants.

## 3 Results of numerical simulations

Our numerical modeling begins from steady state windless non-perturbed atmosphere with profiles of background
temperature, density, molecular weight and molecular kinematic viscosity corresponding to January at latitude 50° N at
medium solar activity according to the NRLMSISE00 model (Picone et al., 2002), which one can find in Figure 1 of the
paper by Gavrilov et al. (2018).

In this study, we consider AGW modes propagating along the eastward axis $x$ and assume the horizontal dimension of
considered atmospheric region to be equal to the circle of latitude at 50°N, which is $L_x \approx$ 27000 km. At horizontal
boundaries of this circle of latitude, we use periodical boundary conditions according to Eq. (6). Representing the circle of
latitude by a rectangle area assumes fixed $L_x$ at all altitudes, while in spherical coordinates $L_x$ is increasing in altitude.
However, the differences in $L_x$ at altitudes of the middle atmosphere do not exceed 2%. Modeling was performed with the
surface wave source (Eq. (5)) for AGW modes having amplitudes $W_0 = 0.01 – 0.1$ mm/s. The smallest amplitudes
correspond to weak AGWs, for which nonlinear effects are small at all considered altitudes. Excitations at $W_0 \sim 0.1$ mm/s
produces stronger AGWs with substantial nonlinear interactions in the mesosphere and lower thermosphere. Used range of
the horizontal phase speed $c_x \sim 50 - 200$ m/s corresponds to AGW modes with relatively large vertical wavelengths, which
are capable to propagate from the ground up to the upper atmosphere. The number of wave periods along the circle of
latitude is taken to be $n_1 = 32$. This corresponds to the horizontal wavelength of $\lambda_x = L_x/n_1 \approx 844$ km and AGW periods of $\tau = \lambda_x/c_x \sim 4.7 – 1.2$ hr for the specified above range of $c_x$ values. The horizontal grid spacing of the numerical model is $\Delta x = \lambda_x/16$ and the time step of calculations was automatically adjusted to $\Delta t \approx 2.9$ s. The vertical grid of the model covers
altitudes up to $h = 600$ km and contains 1024 non-equidistant nodes. Vertical spacing varies between 12 m and 3 km from the
lower to the upper boundary, so about 70% grid nodes are located in the lower and middle atmosphere.

For parameters of the smoothing factor in Eq. (7) in the present simulations, we take $t_a = 10^5$ s $\approx 28$ hr and $t_d = 4 \times 10^5$ s $\approx$
110 hr, and consider gradual AGW source activating and deactivating with $s_a = s_d = 3.3 \times 10^4$ s $\approx 9$ hr and sharp triggering at

$s_a = s_d = 0.3$ s. The shape of the smoothing factor (Eq. (7)) influences the spectrum of the surface wave source in the model. Figure 1 shows spectra of the sinusoidal source (Eq. (5)) with wave period $\tau = 2\pi/\sigma = 2$ hr, which were calculated using 20-
hour running time intervals corresponding to the phases of activation, activated state and deactivation of the wave source (Eq. (5)) with the mentioned above "gradual" and "sharp" values of $s_a$ and $s_d$ in Eq. (7). Comparisons of solid and dashed lines in Figure 1 show that the sharp activation and deactivation of the wave source decreases the spectral density at frequency of the main spectral maximum. However, the sharp triggering considerably increases the high-frequency part of the wave spectra in Figure 1, which means larger proportions of acoustic waves generated by quickly varying wave sources
in the atmosphere.

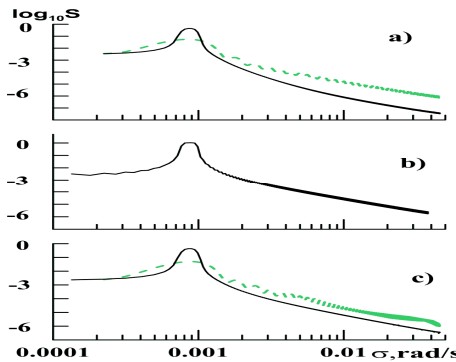

**Figure 1.** Spectral density (in relative units) of the surface wave source (Eq. (5)) having period of 2 hr for 20-hour running time intervals centered at model times $t \approx 20$ hr (a), $t \approx 70$ hr (b) and $t \approx 120$ hr (c), which correspond to the wave forcing activation, activated state and deactivation. Solid and dashed lines correspond to the gradual and sharp
activation/deactivation rates $s_a$ and $s_d$ in (7).

**3.1 Gradual wave source triggering**

Figure 2 shows time variations of the standard deviation of wave vertical velocity $\delta w$ at different altitudes averaged over one horizontal wavelength for the gradual activation and deactivation of the surface wave source (Eq. (5)) with $W_0 = 0.01$ mm/s and $c_x = 50$ m/s. The standard deviation $\delta w$ is proportional to the amplitude of wave variations of vertical velocity. Vertical
dashed lines in Figure 2 show moments $t = t_a \approx 28$ hr and $t = t_d \approx 110$ hr of the surface wave source activation and deactivation in Eq. (7).

The bottom left panel of Figure 2 for the Earth's surface shows that the wave source amplitude increases gradually at $t < t_a$, maintains constant at $t_a < t < t_d$ and gradually decreases to zero at $t > t_d$ in accordance with Eq. (7). Similar increases in $\delta w$ during the activation interval $t < t_a$ one can see at all altitudes in Figure 2. At altitudes higher than 60 km noisy
components are noticeable in Figure 2 at $t < t_a$, which can be produced by acoustic components of the wave source spectrum shown in Figure 1a. However for the gradual smoothing factor $q(t)$ in Eq. (7) this acoustic noise is substantially smaller than the wave amplitudes at $t > t_a$ at all altitudes.

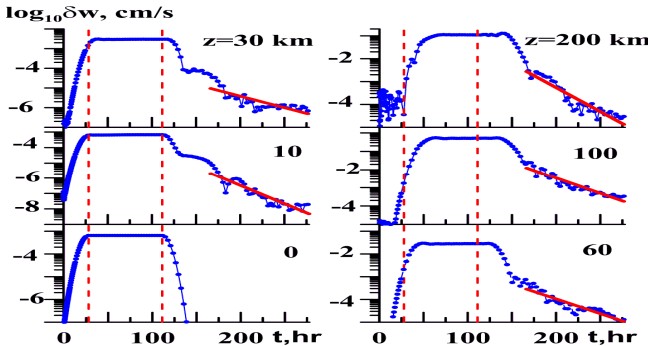

**Figure 2.** Time variations of standard deviations of the wave vertical velocity at different altitudes (marked with numbers) for the gradual activation and deactivation of the surface wave source (Eq. (5)) at $c_x = 50$ m/s and $W_0 = 0.01$ mm/s. Dashed lines correspond to $t = t_a$ and $t = t_d$ in Eq. (7). Solid red lines show exponential fits.

In Figure 2, one can see later transition to a quasi-stationary wave regime with steady amplitudes at higher altitudes compared to that at the Earth's surface. This reflects a time delay $\tau_e \sim z/c_z$ required for the main modes of internal gravity waves (IGWs) to propagate from the surface to altitude $z$ with the mean vertical group velocity $c_z \sim \lambda_z/\tau$, where $\lambda_z$ and $\tau$ are the mean vertical wavelength and wave period, respectively (see Gavrilov and Kshevetskii, 2015). For the shown in Figure 2 wave excitation according to Eq. (5) with $\lambda_x = 844$ km and $c_x = 50$ m/s, using the traditional theory of AGWs (e.g., Gossard and Hooke, 1975), one can estimate $\tau \sim 4.7$ hr, $\lambda_z \sim 15$ km and $t_e \sim (6.7 - 13.3)\tau \sim 31 - 62$ hr for $z = 100 - 200$ km. This corresponds to the time delays between the moments $t_a$ and achieving quasi-stationary amplitudes at different altitudes in Figure 2.

The main goal of this study is the analysis of wave fields, remaining after deactivations of the surface wave sources (Eq. (5)), which we call later as "residual waves". In this section, we applied Eq. (7) with $t_d \approx 110$ hr and $s_d \approx 9$ hr for the gradual wave excitation triggering. Figure 2 shows that after the wave source deactivation, AGW amplitudes start to decrease from their quasi-stationary values at all altitudes with time delays $t_e$ discussed above. Just after the wave forcing deactivation, $\delta w$ decreases relatively fast similarly to the decrease in the wave source amplitude in the bottom left panel of Figure 2. This may reflect disappearing of fast traveling AGW modes due to discontinuities of their generation after the wave forcing deactivation. However, later, at $t > 170$ hr all panels in Figure 2 demonstrate slower $\delta w$ decreases, which can be approximated by exponential curves $\delta w \sim exp(-t/\tau_0)$, where $\tau_0$ is the decay time. Simulations for other values of $c_x$ and $W_0$ showed behavior similar to Figure 2 with differences in the decay time $\tau_0$, which are presented in Table 1 for different altitudes.

For the gradual deactivation of the low-amplitude wave source shown in Figure 2, the decay times in Table 1 are $\tau_0 \sim 17 - 98$ hr depending on altitude, which is much larger than the time scale of the gradual deactivation $s_d \approx 9$ hr. In the middle atmosphere, our model involves the same dissipation mechanisms as at higher altitudes, namely, molecular and turbulent viscosity and heat conduction, also instabilities and nonlinear effects leading to generation of short-wave modes, which we

 call later as "secondary waves". The rate of AGW dissipation depends on the vertical wavelength. Short-wave components may effectively dissipate in the middle atmosphere. However, long-wave modes can propagate up to the upper atmosphere. Slow decay rates shown in Table 1 may be caused by partial reflections of the wave energy resulting in vertically standing AGW modes (see section 4).

**Table 1.** AGW decay times $\tau_0$ in hr in the interval $t \sim 170 - 290$ hr at different altitudes for various parameters of the surface wave sources (Eq. (5)) and their time dependences (Eq. (7)).

| $s_a$, $s_d$, s | $3 \cdot 10^{-1}$ | | | | $3 \cdot 10^{4}$ | | | |
|---|---|---|---|---|---|---|---|---|
| $W_0$, mm/s | 0.01 | | 0.1 | | 0.01 | | 0.1 | |
| $C_x$, m/s | 50 | 100 | 50 | 100 | 50 | 100 | 50 | 100 |
| z = 10 km | 44 | 47 | 54 | 64 | 17 | 54 | 17 | 53 |
| z = 30 km | 67 | 44 | 69 | 46 | 37 | 57 | 34 | 55 |
| z = 60 km | 85 | 85 | 69 | 72 | 33 | 98 | 35 | 92 |
| z = 100 km | 53 | 63 | 52 | 72 | 26 | 60 | 24 | 57 |
| z = 200 km | 54 | 41 | 41 | 40 | 21 | 41 | 61 | 54 |

Contributions may also occur from slow components of the wave source spectrum (see Figure 1), which can dominate after the recession of faster primary spectral modes. In addition, slow shortwave secondary AGW modes can be produced by nonlinear wave interactions at all stages of high-resolution simulations. Mentioned residual and secondary wave modes can slowly travel to higher atmospheric levels and dissipate there due to increased molecular and turbulent viscosity and heat conductivity, which are small in the lower and middle atmosphere. Therefore, decaying these residual and secondary AGW modes may require substantial time intervals after deactivating wave forcing, as one can see in Figure 2.

### 3.2 Sharp wave source triggering

Figure 3 shows the same standard deviation of wave vertical velocity $\delta w$ as Figure 2, but for the sharp activation of the surface wave source according to Eq. (5) with $W_0 = 0.01$ mm/s, $c_x = 50$ m/s and parameters in the time factor of Eq. (7) $t_a \approx$ 28 hr, $t_d \approx 110$ hr and $s_a = s_d = 0.3$ s. The initial AGW pulses are more intensive and contain wider ranges of spectral components (see Figures 1a and 1c) in the case of sharp wave source activations/deactivations. The right top panel of Figure 3 shows that at high altitudes the initial wave pulses might be so high that AGW amplitudes do not reach steady-state conditions existing in the respective panel of Figure 2.

The right top panel of Figure 3 shows substantial AGW pulses not only at the wave source activation $t_a$, but also at the time of wave source sharp deactivation $t_d$, when $\delta w$ values have additional maxima at high altitudes. Stronger AGW pulses caused by the sharp wave source activation/deactivation increase proportions of slow residual and secondary wave components after turning off the wave forcing in the AtmoSym model. Therefore exponential decays of $\delta w$ start earlier and

are more pronounced in Figure 3 than those in the respective panels of Figure 2. AGW decay times $\tau_0$ corresponding to the exponential approximations in Figure 3 for the sharp wave source activation are given in the left column of Table 1 and vary between 44 and 85 days. They are generally larger than the discussed above values of $\tau_0$ in Figure 2, which means that stronger residual wave noise for the case of sharp wave source triggering require longer time intervals for their decay.

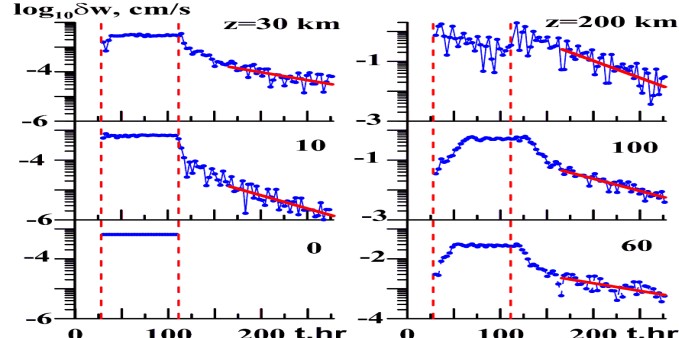

**Figure 3.** Same as Figure 2, but for the sharp wave source activation.

Figures 2 and 3 represent results for the wave source (Eq. (5)) with $c_x = 50$ m/s. Table 1 contains also the decay times for the wave excitation with $c_x = 100$ m/s. Respective primary AGWs have larger vertical wavelengths and should experience smaller molecular and turbulent dissipation in the atmosphere. For the gradual activation/deactivation of the wave source (Eq. (5)) with small amplitude $W_0 = 0.01$ mm/s, Table 1 reveals larger values of $\tau_0$ for AGWs with $c_x = 100$ m/s compared to those with $c_x = 50$ m/s. Therefore, smaller dissipation of the faster AGW modes corresponds to longer time for their decay, especially at altitudes below 100 km. For the sharp activation/deactivation of the wave source at $W_0 = 0.01$ mm/s, the left columns of Table 1 shows approximately equal $\tau_0$ values for waves with $c_x = 50$ m/s and $c_x = 100$ m/s.

**Table 2.** Ratios $\delta w(z)/W_0$ at $t = 170$ hr at different altitudes for various parameters of the surface wave source (Eq. (5)) and its time dependence according to Eq. (7).

| $s_a, s_d,$ s | $3 \cdot 10^{-1}$ | | | | $3 \cdot 10^4$ | | | |
|---|---|---|---|---|---|---|---|---|
| $W_0$, mm/s | 0.01 | | 0.1 | | 0.01 | | 0.1 | |
| $c_x$, m/s | 50 | 100 | 50 | 100 | 50 | 100 | 50 | 100 |
| z = 10 km | 0.015 | 0.016 | 0.013 | 0.013 | 0.003 | 0.001 | 0.003 | 0.001 |
| z = 30 km | 0.158 | 0.163 | 0.182 | 0.190 | 0.009 | 0.001 | 0.011 | 0.001 |
| z = 60 km | 2.272 | 1.032 | 2.257 | 1.147 | 0.289 | 0.004 | 0.257 | 0.005 |
| z =100 km | 44.58 | 12.88 | 42.36 | 11.97 | 12.39 | 0.086 | 10.55 | 0.082 |
| z =200 km | 198.3 | 335.8 | 277.5 | 344.7 | 2.732 | 0.925 | 0.426 | 0.406 |

Relative contributions of residual and secondary AGWs can be estimated by the ratio $\delta w/W_0$ at the beginning of the exponential tails in Figures 2 and 3 at $t = 170$ hr, which is presented in Table 2. For the gradual wave source

activation/deactivation at $s_a = s_d \approx 9$ hr in Eq. (7) and $W_0 = 0.01$ mm/s in Eq. (5), Table 2 shows smaller ratios of the residual wave noise at altitudes below 200 km for the wave forcing with $c_x = 100$ m/s compared to $c_x = 50$ m/s. At the sharp wave source activation/deactivation at $s_a = s_d \approx 0.3$ s in Eq. (7), the ratios of residual waves are larger at all altitudes compared to the gradual case in Table 2. For the wave forcing (Eq. (5)) with $c_x = 100$ m/s, the ratios are comparable or smaller at altitudes below 150 km and larger above 150 km compared to the case of $c_x = 50$ m/s. Larger ratios of residual and secondary waves at sharp wave source triggering in Table 2 may explain generally larger AGW decay times $\tau_0$ in the left columns of Table 1 for $W_0 = 0.01$ mm/s, as far as the dissipation of stronger wave noise may require longer time intervals.

### 3.3 Larger amplitude wave sources

Described above simulations were made for small-amplitude wave sources (Eq. (5)) with $W_0 = 0.01$ mm/s. For larger $W_0 = 0.1$ mm/s, Figure 4 reveals time variations of the vertical velocity standard deviations $\delta w$ at different altitudes for $c_x = 100$ m/s at the gradual wave source activation/deactivation with $s_a = s_d \approx 9$ hr s in Eq. (7), which is similar to Figure 2.

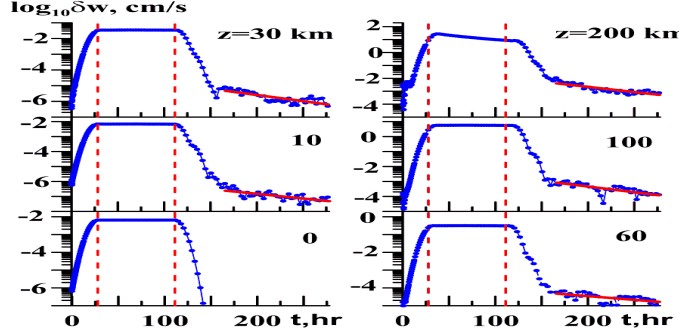

**Figure 4.** Same as Figure 2, but for the surface wave source (Eq. (5)) at $c_x = 100$ m/s and $W_0 = 0.1$ mm/s.

Below altitude of 100 km, one can see the intervals of quasi-constant AGW amplitudes after the end of the wave source activation at $t = t_a$ (vertical dashed lines in Figure 4). Theoretical time delay $t_e$ between the wave source activation and the beginning of the steady-state AGW regime is 4 times smaller for $c_x = 100$ km than that for $c_x = 50$ km, as one can see comparing Figures 2 and 4. After deactivations of the surface wave source (Eq. (5)) at $t = t_d$, values of $\delta w$ in Figure 4 are first decreasing relatively fast due to the discontinuing generation of primary AGW modes at the lower boundary. At $t > 150 – 170$ hr, more slow decays of residual and secondary wave modes occur at all altitudes in Figure 4 with decay times $\tau_0$ listed in Table 1 for the gradual and sharp wave source activation/deactivation.

Peculiarities of Figure 4 for large $W_0$ are gradual decreases in AGW amplitudes during the wave source operation between moments $t_a$ and $t_d$ at high altitudes (see the top right panel of Figure 4) in comparison with steady amplitudes in respective panels of Figure 2 for smaller $W_0$. The reason could be strong generations of wave-induced jet streams by large-amplitude AGWs at high altitudes. Figure 5 shows time variations of horizontal velocity $u_0$ averaged over a period of the surface wave source (Eq. (5)) with $W_0 = 0.1$ mm/s at different altitudes.

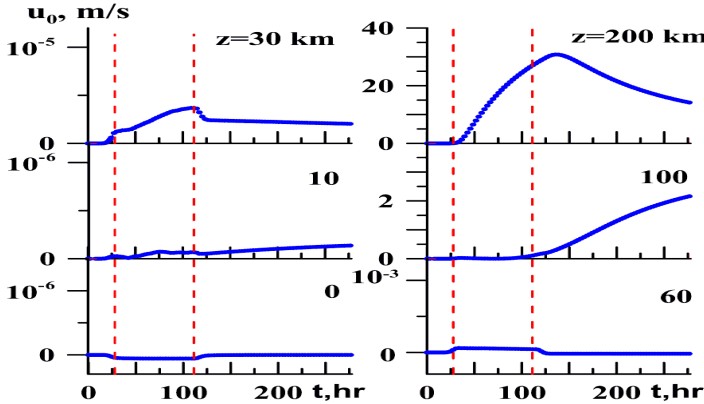

**Figure 5.** Time variations of the wave induced mean horizontal velocity at different altitudes (marked with numbers) for the gradual activations and deactivations of the surface wave source (Eq. (5)) at $c_x = 100$ m/s and $W_0 = 0.1$ mm/s. Dashed lines correspond to $t = t_a$ and $t = t_d$ in Eq. (7).


Generation of the wave-induced jet streams was simulated and considered in more details in our previous papers (Gavrilov and Kshevetskii, 2015; Gavrilov et al., 2018). Larsen (2000) and Larsen et al. (2005) found frequent high horizontal wind velocities at altitudes near 100 km, which could be related to the wave-induced jet streams. In Figure 5 for the strong wave excitation with amplitude of $W_0 = 0.1$ mm/s, one can see substantial $u_0$ rises at altitudes above 100 km during the wave

source operation. Rising $u_0$ decreases the AGW intrinsic frequency and vertical wavelength (e.g., Gossard and Hooke, 1975). This may increase wave dissipation due to molecular viscosity and heat conductivity leading to the gradual decrease in AGW amplitude in the right top panel of Figure 4 in the time interval between $t_a$ and $t_d$. The rate of $u_0$ weakening after the wave source deactivation decreases slowly in time in the right top panel of Figure 5, so that the wave-induced horizontal winds are still substantial after hundreds wave source periods at high altitudes. An interesting feature is an increase in $u_0$ at $t > t_d$ in the

panel of Figure 5 for $z = 100$ km. This shows that residual and secondary AGWs slowly traveling upwards from below can produce substantial wave accelerations of the mean flow for long time after deactivations of the surface wave sources.

     Table 2 represents the ratio $\delta w/W_0$ at the moment $t \approx 170$ hr for larger-amplitude surface wave sources (Eq.(5)), which may characterize a proportion of residual and secondary waves after disappearing the fast traveling modes of the wave excitation. At gradual wave source activations/deactivations with $s_a = s_d \approx 9$ hr, Table 2 demonstrates approximately same

$\delta w/W_0$ values below altitude of 100 km and generally smaller values at higher altitudes for $W_0 = 0.1$ mm/s as compared with $W_0 = 0.01$ mm/s, if one considers columns for fixed $c_x$ at different $W_0$ values. This may be caused by the discussed above transfer of wave energy to wave-induced jets, which can provide also larger reflection and dissipation of wave components with larger amplitudes.

     AGW decay times in Table 1 for $W_0 = 0.1$ mm/s at altitudes below 100 km are generally larger for the sharp wave source

triggering ($s_a = s_d \approx 0.3$ s) than those for the gradual triggering ($s_a = s_d \approx 9$ hr) similar to the case of smaller wave source

amplitude discussed in section 3.2. At high altitudes in Table 1 for $W_0 = 0.1$ mm/s, wave decay times for the sharp wave source deactivating become smaller, than those for the gradual triggering.

### 3.4 Spatial structure of AGW fields

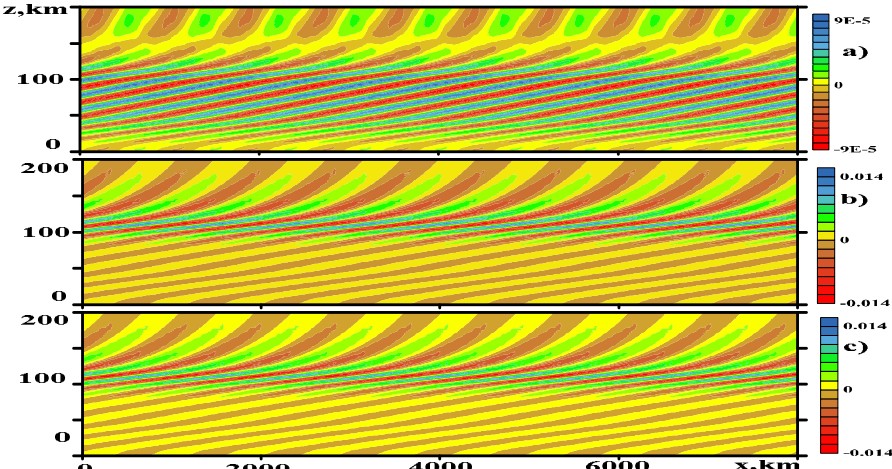

**Figure 6.** AGW vertical velocity fields at times $t \approx 30$ hr (a), $t \approx 70$ hr (b) and $t \approx 110$ hr (c) for the small rate of gradually activating/deactivating the wave source (Eq. (5)) with $c_x = 50$ m/s and $W_0 = 0.01$ mm/s.

To analyze changes in the spatial structure of simulated AGW fields, Figures 6 and 7 present cross-sections of the field of wave vertical velocity by a vertical plane at different time moments during activations and deactivations of the surface wave
sources (Eq. (5)) with the gradual values of $s_a = s_d \approx 9$ hr in Eq. (7). Figure 6a shows that after dispersion and dissipation of the initial AGW pulse just after the wave forcing activation time, $t_a \approx 28$ hr, wave fronts become inclined to the horizon. This behavior is characteristic for the main IGW mode with period $\tau \sim 4.7$ hr, which is dominating in the spectrum of the wave source having $c_x = 50$ m/s similar to Figure 1a. In the middle and at the end of quasi-stationary intervals shown in Figures 2 – 4, the inclined wave fronts in Figures 6b and 6c expand to the entire considered atmospheric region and wave amplitudes
become larger compared to Figure 6a.

Cross-sections shown in Figure 7 correspond to time moments after the wave source deactivation at $t_d \approx 110$ hr. Figure 7a shows that just after turning off the wave source, the inclined fronts are destroyed, first, in the lower atmosphere. Above altitude 50 km, the wave field structure in Figure 7a is still similar to Figures 6b and 6c. Later, in Figures 7b and 7c, wave amplitudes become smaller, especially at low and high altitudes. Therefore, maximum AGW amplitudes in Figure 7c are
located at altitudes 80 – 120 km. This explains the growing wave-induced horizontal velocity at altitude 100 km after the wave source deactivation in the respective panel of Figure 5. At heights below 50 km in Figure 7, directions of wave front inclinations to the horizon are opposite to those in Figure 6. This reveals existence of downward traveling IGW modes in the

stratosphere and troposphere after deactivations of the surface wave sources. Such modes could be produced by partial reflections of primary upward traveling IGWs at higher atmospheric levels (see section 4).

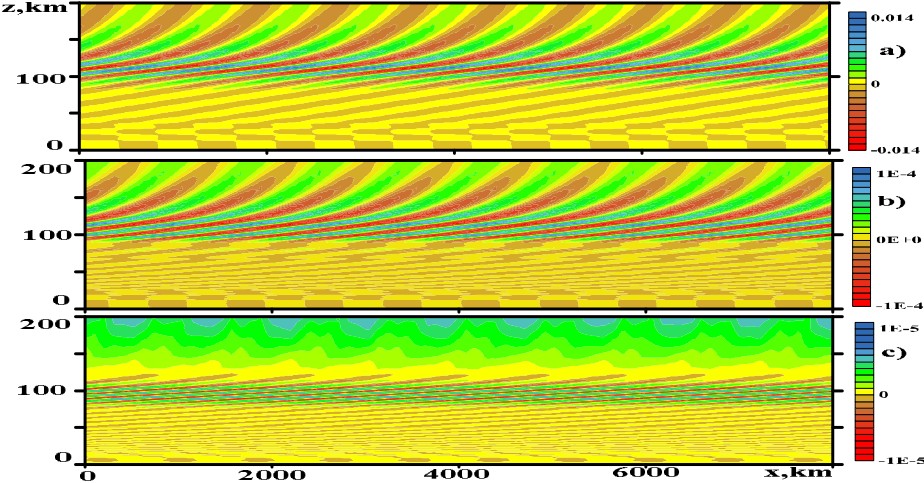

**Figure 7.** Same as Figure 6, but for time moments after the wave source gradual deactivating: $t \approx 140$ hr (a), $t \approx 200$ hr (b) and $t \approx 250$ hr (c).

Figures 7b and 7c show increasing amounts of small-scale structures, which can be formed by slow shortwave residual wave modes, which appear due to broad wave source spectra in Figure 1 and due to generating secondary waves by nonlinear interactions of primary AGW modes.

## 4 Discussion

The time scale of AGW dissipation in the turbulent atmosphere can be estimated as follows (Gossard and Hooke, 1975):

$$\tau_d = \frac{\lambda_z^2}{2\pi K_z}, \tag{8}$$

where $K_z$ is the total vertical coefficient of turbulent and molecular viscosity and heat conductivity. For the main primary AGW modes simulated in this study and having $\lambda_z \sim 15 - 30$ km (see section 3.1), $\tau_d \sim 10^3 - 10^5$ hr at altitudes below 100 km. These values are much larger than the AGW decay times $\tau_0$ in Table 1. Therefore, attenuations of primary AGW modes in the middle atmosphere shown in Figures 2 -7 after deactivations of the surface wave forcing cannot be explained by direct turbulent and molecular dissipations.

AGWs propagating in the atmosphere with vertical gradients of the background fields are subjects to partial reflections. In particular, strong wave reflections occur at altitudes 110 – 150 km, where large vertical gradients of the mean temperature exist (e.g., Yiğit and Medvedev, 2010; Walterscheid and Hickey, 2011; Gavrilov and Kshevetskii, 2018). Partial reflections of wave energy propagating upwards from the wave sources before their deactivations may produce vertically standing waves in the middle atmosphere. Simulations by Gavrilov and Yudin (1987) showed that the standing-wave ratio for IGW

amplitudes might reach 0.4 at altitudes below 100 km. After deactivations of wave sources, vertically traveling AGW modes propagate quickly upwards and dissipate at higher atmospheric altitudes. This gives fast decreases in AGW amplitudes at all heights in Figures 2 and 4 just after the wave source deactivations. After disappearing fast traveling modes, residual vertically standing AGWs produced by partial reflections may form long-lived wave structures in the atmosphere shown in Figures 2 – 7.

The standing AGWs discussed above are composed of the primary wave modes traveling upwards from the surface wave sources (Eq. (5)) and downward propagating waves reflected at higher atmospheric levels. After the wave source deactivations, the reflected downward waves may propagate to the Earth's surface and create wave fronts at low altitudes in Figure 7, which are inclined to the horizon in directions opposite to the fronts of primary AGWs shown in Figure 6. For used smooth climatological temperature profiles from the NRLMSISE-00 model (see Figure 1 of the paper by Gavrilov et al., 2018) AGW reflections inside the troposphere are smaller than the reflection from the ground caused by lower boundary conditions given by Eq. (4). Therefore, mentioned above downward traveling waves are reflected from the ground and propagate upwards back to the middle atmosphere. Kurdyaeva et al. (2018) showed that such AGW reflections from the ground could be equivalent to additional wave forcing at the lower boundary, which is still effective after deactivations of primary surface wave sources. Upward traveling from the ground and reflected again at higher altitudes waves can maintain standing AGW structures for long time (see Figure 7). As far as wave reflections are partial, portions of wave energy can for long time propagate to higher altitudes and dissipate there. This can explain relatively large AGW decay times $\tau_0$ in the lower and middle atmosphere shown in Figures 2 – 4 and in Table. 1. Even after substantial time from the wave source turning off, AGW structures in Figures 7b and 7c at altitudes above 50 km are still similar to those shown in Figure 6 during active wave forcing.

Panels of Figure 2 for the gradual wave source activation/deactivation demonstrate periodical increases and decreases in the residual wave noise standard deviations (especially at low altitudes), which are superimposed on the exponential decay at $t > t_d$. This may be caused by long-term biases between upward and downward wave packages reflected from the ground and from the upper atmosphere, which propagate through the middle atmosphere. Increased molecular and turbulent AGW dissipation make periodical amplitude variations less noticeable in the panels of Figure 2 for high altitudes. These biases are also less noticeable in the respective panels of Figure 3 for the sharp wave source activation, because the wave source spectra in Figure 1 are smoother and wider in this case compared to the smooth deactivation of the wave excitation.

One can rise a question to what extend the results shown in Tables 1 and 2 may depend on so-called "numerical viscosity" caused by mathematical algorithms used in the model? Our model is based on special numerical algorithms accounting for the main conservation laws (Gavrilov and Kshevetskii, 2013, 2014). Therefore, the numerical viscosity is very small. Test simulations showed that in the absence of physical dissipation, wave modes might exist in the model for hundreds of wave periods without noticeable decreases in their amplitudes. In addition, simulated ratios of standard deviations of different components of long-wave fields in the middle atmosphere follow to the polarization relations of conventional theory of nondissipative AGWs (Gavrilov et al., 2015). Therefore, we assume that in the present model, the

numerical viscosity is much smaller than the molecular and turbulent viscosity and heat conduction, which are involved in the model at all altitudes.

Shown in Table 2 ratios $\delta w/W_0$ at $t \approx 170$ km may reflect proportions of the residual and secondary AGW modes in the beginning of quasi-exponential fits in Figures 2 – 4. For the gradual wave forcing activations/deactivations, in the right part of Table 2, one can see larger ratios for wave modes with $c_x = 50$ m/s at all altitudes. This corresponds to longer intervals of

fast decreases of AGW amplitudes after deactivations of the wave sources in Figure 4 compared to Figure 2. Considerations of respective right columns of Table 1 reveal larger decay times $\tau_0$ of waves with $c_x = 100$ m/s due to their larger vertical wavelength and smaller dissipation in the middle atmosphere.

Comparisons of the right columns in Table 2 with the same $c_x$ and different $W_0$ show that values of $\delta w/W_0$ for each $c_x$ are approximately equal at altitudes below 60 km and become smaller at higher altitudes for larger amplitude wave sources. This

may reflect larger transfers of AGW energy to wave-induced jet streams and to secondary nonlinear modes produced by larger-amplitude waves. Respective right columns of Table 1 show higher decay times $\tau_0$ of larger-amplitude wave noise corresponding to $W_0 = 0.1$ mm/s at altitudes higher 100 km. This noise can be maintained for long time by wave energy fluxes propagating with stronger residual and secondary waves from the middle atmosphere to higher altitudes.

For the sharp activations/deactivations of the wave sources (Eq. (5)), the left columns of Table 2 show values of $\delta w/W_0$,

which are much larger compared to respective right columns for the gradual wave forcing triggering. These ratios are less dependent on the speed and amplitude of simulated AGWs and could be connected with wave pulses produced by sharp activations/deactivations of the wave sources (see spectra in Figure 1). AGW decay times for the sharp triggering in respective left columns of Table 1 are also less dependent on wave parameters.

Substantial amount of small-scale structures in Figures 7b and 7c shows increased proportions of wave modes, produced

due to high-frequency tails of the wave forcing spectra in Figure 1, also due to multiple reflections and nonlinear interactions of these modes. Nonlinear AGW interactions and generations of secondary waves should be stronger at high altitudes due to increased wave amplitudes (Vadas and Liu, 2013; Gavrilov at al., 2015). Then the secondary waves can propagate downwards and make small-scale wave perturbations at all atmospheric altitudes (see Figures 6 and 7). The AGW decay times $\tau_0$ in Table 1 are generally larger for longer AGW modes with $c_x = 100$ m/s. This may be explained by their smaller

dissipation due to turbulent and molecular viscosity and heat conductivity in the atmosphere. Due to small coefficients of turbulent dissipation in the stratosphere and mesosphere, maximum AGW decay times in Table 1 exist at altitudes 30 – 100 km. Standing and secondary AGWs may exist there for several days after deactivations of the wave forcing. Wave energy can slowly penetrate upwards from the stratosphere and mesosphere and maintain a background level of AGW activity at higher altitudes. Figure 7c reveals that after 10 days of simulations, largest amplitudes of the residual wave field exist at

altitudes 70 – 110 km. It is enough for creations of wave accelerations, which can act and modify the mean velocity at altitudes near 100 km for the long time after the wave source deactivations (see respective panels of Figure 5).

Simulations presented in this paper are made for horizontally uniform wave excitation at the ground described by Eq. (5). At the same time, many wave sources are localized in different atmospheric regions. Our test simulations for localized

wave sources (e.g., Kurdyaeva et al., 2018) showed that near an isolated deactivated wave source the amplitude decay could be faster due to horizontal dispersion of wave packets. However, at low altitudes these wave packets can several times go around the globe and return to the initial point similar to wave packages observed after big explosions of meteorites and volcanoes (e.g., Ewing and Press, 1955; Roberts et al., 1982). Therefore, globally, wave packets may exist in the atmosphere for a long time. For several local wave sources, wave packets from different sources may overlap and produce more horizontally uniform long-lived wave noise. Therefore, the horizontally inhomogeneous model considered in this paper may reflect general global features of AGW decay processes in the atmosphere. Studies of isolated and multiple local wave sources require special considerations in subsequent papers.

Described above simulations were made for single relatively long AGW spectral components, which experience small dissipation in the stratosphere and mesosphere. Real wave fields in the atmosphere are superpositions of wide range of spectral components generated by a variety of different wave sources. However, after deactivations of wave sources, fast traveling spectral components disperse to higher altitudes and short wave modes are strongly dissipated due to turbulent and molecular viscosity and heat conductivity. Therefore, one may expect that at the final stage of wave disappearing after deactivations of wave forcing, wave fields in the stratosphere and mesosphere should consist of vertically standing relatively long spectral components, similar to those considered in the present study. These wave fields may contain substantial proportions of residual and secondary wave modes produced by multiple reflections and nonlinear interactions. Such impression is probably true for the residual wave noise, which may exist for long time after the wave source deactivation. However, amplitudes of this residual noise become smaller in time and near active wave sources, amplitudes of generated primary AGWs may much exceed the wave noise.

In this paper, we analyzed idealistic cases of long-lived horizontally homogeneous coherent wave sources producing quasi-stationary wave fields in the atmosphere. Such modeling is useful for comparisons of simulated results with standard AGW theories. However, many AGW sources in the atmosphere are local and operate for short time, which is not enough for developments of steady-state wave fields. Further simulations are required for studying wave decay processes after deactivating such local short-lived wave sources in the atmosphere.

## 5 Conclusion

In this study, the high-resolution numerical model AtmoSym is applied for simulating non-stationary nonlinear AGWs propagating from surface wave sources to higher atmospheric altitudes. After activating the surface wave forcing and fading away initial wave pulses, AGW amplitudes reach a quasi-stationary state. Then the surface wave forcing is deactivated in the numerical model and amplitudes of primary traveling AGW modes quickly decrease at all altitudes due to discontinuation of wave energy generation by the surface wave sources. However, later the standard deviation of the residual and secondary wave perturbations produced by slow components of the wave source spectrum, multiple reflections and nonlinear interactions experiences more slow exponential decreases. The decay time of the residual AGW noise may vary between 20

and 100 hr, having maxima in the stratosphere and mesosphere. Standard deviations of the residual AGWs in the atmosphere are much larger at sharp activations/deactivations of the wave forcing compared to the gradual processes. These results show that transient wave sources in the lower atmosphere could create long-lived residual and secondary wave perturbations in the middle atmosphere, which can slowly propagate to higher altitudes and form a background level of wave noise for time

intervals of several days after deactivations of wave sources. Such behavior should be taken into account in parameterizations of AGW impacts in numerical models of dynamics and energy of the middle atmosphere.

**Data availability.** Used high-resolution model of nonlinear AGWs in the atmosphere is available for online simulations (see the reference AtmoSym, 2017). The computer code can be also available under the request from the authors.


**Author contributions.** SPK participated in the computer code development. AVK prepared background fields for the simulations. NMG made simulations and prepared the initial text of paper, which was edited by all authors.

**Competing interests.** The authors declare that they have no conflicts of interests.


**Acknowledgments.** AGW numerical simulations were made in the SPbU Laboratory of ozone layer and upper atmosphere supported by the Ministry of Science and High Education of the Russian Federation (agreement 075-15-2021-583**).**

**Financial support.** Calculations of the background fields for simulations were supported by the Russian Science Foundation

(grant 20-77-10006).

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
