# Peer review of "Decay times of atmospheric acoustic-gravity waves after deactivation of wave forcing"

_Atmospheric Chemistry and Physics, 2021_

## Author Response (AR2)

**Reply to the Reviewer 1**

of the paper "Decay times of atmospheric acoustic-gravity waves after deactivation of wave forcing" by N.M. Gavrilov et al.

**We would like to thank the Reviewer for interesting and valuable comments helping us to improve the paper. Our replies are given below in the bold font.**

This paper presents a series of numerical simulations with a high-resolution two-dimensional wave model addressing responses of acoustic-gravity waves to transient sources. The latter were gradually activated and deactivated at the lower boundary using two characteristic time scale scenarios. While modelers are quite familiar with the "growing" phase, the behavior of the wave field after sources are turned off was least studied. The paper deserves publication, however the presentation has to be improved.

Some general points are listed below.
1. Some discussion is required on what mechanisms (beside the numerical viscosity) cause wave dissipation in the lower and middle atmosphere. It seems that molecular diffusion in the thermosphere is the major mechanism that eliminates waves in the modeling domain.

**In the middle atmosphere, our model involves the same dissipation mechanisms as at higher altitudes, namely, molecular and turbulent viscosity and heat conduction, also instabilities and nonlinear effects leading to generation of secondary short-wave modes. The rate of AGW dissipation depends on the wavelength. Short-wave components may effectively dissipate in the middle atmosphere. Long-wave modes can propagate up to the upper atmosphere. The model involves all these cases. We add more clarification to the revised text of the paper.**

2. The results are essentially summarized in two tables. How numerical viscosity affects these numbers? Are they representative, or just depend on the particular model?

**The model is based on special numerical algorithms accounting for the main conservation laws. Therefore, "numerical viscosity" is very small. Our test simulations showed that in the absence of physical dissipation, wave modes might exist in the model for hundreds of wave periods without visible decrease in their amplitudes. Therefore, we think that in the present model, viscous dissipation is much smaller than molecular and turbulent viscosity and heat conduction, which are involved in the model at all altitudes. We added respective description to the revised text.**

3. Is there any significance of your results for waves in 3-dimensional case? The decay would apparently be much faster in 3D due to dispersion and localized sources.

**In principle, we have performed some test simulations for localized wave sources. The reviewer is right that locally, for an isolated wave source, the amplitude decay could be faster due to horizontal dispersion of wave packets. However, at low altitudes these wave packets can several times go around the globe and return to the initial point (such behavior was observed, for example, for strong AGWs after big explosions of meteorites and volcanoes). Therefore, globally, wave packets may exist in the atmosphere for the long time. If there are several local wave sources, wave packets from different sources may superpose and produce more horizontally uniform long-lived wave noise. Therefore, the horizontally inhomogeneous model considered in this paper may reflect general global features of AGW decay processes in the atmosphere. Considerations of isolated and multiple local wave sources we are planning to make in subsequent papers. We added this discussion to the revised text.**

4. Are there implications for the real atmosphere? My impression after reading the manuscript is that the atmosphere is full of run-away packets and individual spectral harmonics.

**Such impression is probably true for the residual wave noise, which may exist for long time after the wave source deactivation. However, amplitudes of this residual noise become smaller in time and near active wave sources, amplitudes of generated primary AGWs may much exceed the wave noise. We added this discussion to the revised text.**

More specific comments
- The terms "quasi-standing", "residual" and "secondary" waves are used, but their meaning is not defined and is not clear (e.g., l. 210).

**"Secondary waves" is a usual term for smaller-scale wave modes generated by forced "primary" waves due to instabilities and nonlinear interactions (e.g. , Healy et al., https://doi.org/10.1029/2019JD031662). The term "residual waves" we use to indicate wave modes propagating in the atmosphere after deactivations of their sources. We added these definitions to the revised text. The term "quasi-standing" is less defined. We changed respective phrases to avoid its usage.**

- l. 180 "This may reflect disappearing of fast traveling AGW modes". - It probably is quite opposite. Deactivating the forcing below introduces a spectrum of harmonics, including those traveling fast.

**Thanks. We made this description more precise.**

- l. 240. "Theoretical time delay t_e..." - Was it introduced/calculated somewhere?

**Yes. The time delay was estimated by Gavrilov and Kshevetskii (https://doi.org/10.1016/j.asr.2015.01.033). We added the citation to the text.**

**Yours sincerely. Nikolai M. Gavrilov, Sergey P. Kshevetskii, and Andrey V. Koval**

**Reply to the Reviewer 2**
of the paper "Decay times of atmospheric acoustic-gravity waves after deactivation of wave forcing" by N.M. Gavrilov et al.

**First, we would like to thank the Reviewer for valuable comments helping us to improve the paper. Our replies are given below in the bold font.**

This paper is a numerical study that investigates the effects that happen after a source of acoustic-gravity waves (AGWs) is deactivated. One of the main findings is that after source deactivation there is a significant amount of "wave noise" that slowly decays quasi-exponentially. The wave noise is attributed to quasi-standing and secondary AGW spectral components. This effect should contribute to the background level of AGW activity in the real atmosphere and is so far neglected in parameterizations of AGWs.

The paper provides several interesting results and is of relevance for the readership of ACP. The paper is well written and recommended for publication in ACP after minor revisions.

Minor comments:

(1) l.23: The paper by Fritts and Alexander (2003) starts with a general dispersion relation, but does not explicitly treat acoustic GWs (AGWs).Therefore it should not be used as an evidence

for the statement that AGWs "exist almost permanently in the atmosphere". I would suggest to replace this reference with other examples of observations and modeling. Some suggestions:

**We added suggested citations into the revised text prepared after the public discussion.**

Lay, E. H. (2018). Ionospheric irregularities and acoustic/gravity wave activity above low-latitude thunderstorms. Geophysical Research Letters, 45, 90-97. https://doi.org/10.1002/2017GL076058.

Meng, X., Vergados, P., Komjathy, A., & Verkhoglyadova, O. (2019). Upper atmospheric responses to surface disturbances: An observational perspective. Radio Science, 54, 1076-1098. https://doi.org/10.1029/ 2019RS006858

Siefring, C. L., J. S. Morrill, D. D. Sentman, and M. J. Heavner (2010), Simultaneous near-infrared and visible observations of sprites and acoustic-gravity waves during the EXL98 campaign, J. Geophys. Res., 115, A00E57, doi:10.1029/2009JA014862.

Snively, J. B. (2013), Mesospheric hydroxyl airglow signatures of acoustic and gravity waves generated by transient tropospheric forcing, Geophys. Res. Lett., 40, 4533-4537, doi:10.1002/grl.50886.

Trinh, Q. T., Ern, M., Doornbos, E., Preusse, P., and Riese, M.: Satellite observations of middle atmosphere-thermosphere vertical coupling by gravity waves, Ann. Geophys., 36, 425-444, https://doi.org/10.5194/angeo-36-425-2018, 2018.

Wei, C., Buehler, O., and Tabak, E. G.: Evolution of Tsunami-Induced Internal Acoustic-Gravity Waves, J. Atmos. Sci., 72, 2303-2317, doi:10.1175/JAS-D-14-0179.1, 2015.

(2) In Fig.2 it is noteworthy that there seems to be a cascade of amplitude decay before the exponential decrease sets in. Particularly at 10km and 30km, there is a fast decrease after wave source deactivation to an intermediate level (between 125 h and 175 h), followed by another fast decrease to a level from where the quasi-exponential decay starts. Do you have any idea what causes this cascade?

**In fact, in Fig. 2 the curves of amplitude decay looks like sinusoidal structures superimposed on exponential trends. We think that such quasi-periodical amplitude variations can be caused by long-term biases between upward and downward wave packages reflected from the ground and from the upper atmosphere, which propagate through the middle atmosphere. Increased molecular and turbulent AGW dissipation make periodical amplitude variations less noticeable in panels of Fig. 2 for high altitudes. We added this statement into the discussion section of the revised text.**

(3) About wave launch amplitudes and phase speed: are the values assumed for your simulations realistic for known source processes?

**Intensity and spectra of atmospheric AGWs are very variable. We made modeling of AGW spectral components for broad ranges of wave source amplitudes. The small amplitude of $W_0$=0.01 mm/s corresponds to weak AGWs, and $W_0$=0.1 mm/s is for rather strong AGWs. The latter are subjects for substantial nonlinear effects. Fig. 2 – 4 and tables 1 and 2 show that general AGW behavior after the wave source deactivating does not depend on amplitudes and phase speeds (differences are in some numerical characteristics only). We added this discussion to the revised text.**

(4) l.254: about "generation of wave-induced jet streams..." Do you think that this can be an important effect in the real atmosphere? Do you think that launch amplitudes are realistic, or could they be too strong and cause this effect?

**For strong AGW sources, wave amplitudes can also be strong and wave-induced jets can be produced at high altitudes. Generation of the wave-induced jet streams was studied in**

**more details in our previous papers (Gavrilov and Kshevetskii, doi:10.1016/j.asr.2015.01.033; Gavrilov et al., 2018). In these papers, we also consider experimental evidences of wave-induced jets in the upper atmosphere. We added these references into the revised text.**

(5) Could there be reflections at the tropopause level in the model and in the real atmosphere? Is the sharp feature of the real-atmosphere tropopause captured in the assumed background atmosphere? Could you show the temperature profile that you use? What would happen if the background wind changes rapidly with height?

**We use smooth temperature profiles from the NRLMSISE-00 model. Our profiles are published in the paper by Gavrilov et al. (2018). For these climatological profiles AGW reflections inside the troposphere are smaller than the reflection from the ground caused by lower boundary condition w'=0. In special cases of strong vertical gradients of background temperature and mean wind AGW reflections in the troposphere could be stronger, however we do not consider such special cases in the present paper. We added respective statements to the revised text.**

Technical comments:
    l.26: AGWs are permanently existed -> AGWs permanently exist.  **Corrected in the revised text.**
    l.26/27 reference Yigit et al. (2012) is missing in the references. Yigit, E., Ridley, A.J., Moldwin, M.B.: Importance of capturing heliospheric variability for studies of thermospheric vertical winds, J. Geophys. Res. 117, A07306. https://doi.org/10.1029/2012JA017596, 2012. **The reference for Yigit et al. (2012) is added.**
    l.29: Gossard and Hook -> Gossard and Hooke **Corrected.**
    l.36: 1916 -> 2016  **Corrected.**
    l.40: there analysis. -> their analysis  ??  **Corrected.**
    l.43: (RAMS ) -> (RAMS)  **Corrected.**
    1.48: propagations ->  propagation **Corrected**
    l.54: of numerical model, ->  of the numerical model, **Corrected**
    l.83: accompanied AGW propagation. -> that accompany AGW propagation. **Corrected**
    l.85: deviations (2) -> deviations as defined in Eq. (2)   **Corrected.**
    l.87:  (Picone et al., 2001). ->  (Picone et al., 2002). **Corrected.**
    l.91:  maxima about ->  maxima of about  **Corrected.**
    l.92: minimum up -> minimum of up   **Corrected.**
    l.98: conditions at the upper boundary (4) -> conditions at the upper boundary as defined in Eq. (4) **Corrected.**
    l.99: conditions (4) -> conditions (Eq. (4))  **Corrected.**
    l.102: ?? have sense of the amplitude and frequency, -> are the amplitude and frequency of wave excitation, **Corrected.**
    l.104:  in (5) ->  in Eq. (5) **Corrected.**
    l.112:  the wave excitation (5) ->  the wave excitation in Eq. (5)  **Corrected.**
    l.113:  of wave source (5), ->  of the wave source in Eq. (5), **Corrected.**
    l.118: the wave source (5)  -> the wave source in Eq. (5)   **Corrected.**
    l.119:  activating surface wave ->  activating of the surface wave  **Corrected.**
    l.120: (5) -> (Eq. (5))  **Corrected.**
    l.121: in (5) -> in Eq. (5) **Corrected.**
    general comment: please check how equations should be referenced in the text according to ACP style!  **The rules are checked.**
    l.127: 2001 -> 2002  **Corrected.**
    l.129: and use horizontal -> and assume the horizontal  **Corrected.**
    l.131:  conditions (6). -> conditions according to Eq. (6). **Corrected.**

l.133: (5) -> (Eq. (5)) **Corrected.**

l.135: This corresponds the horizontal wavelength -> This corresponds to the horizontal wavelength of **Corrected.**

l.136: periods -> periods of **Corrected.**

l.142: of smoothing factor -> of the smoothing factor **Corrected.**

l.174: Hook -> Hooke **Corrected.**

l.178: made applying (7) -> performed by applying Eq. (7) **Corrected.**

l.191: making vertically quasi-standing AGW modes -> resulting in vertically quasi-standing AGW modes **Corrected.**

l.192: of wave source spectrum -> of the wave source spectrum **Corrected.**

l.216: wave noise at the sharp wave source triggering -> wave noise for the case of sharp wave source triggering **Corrected.**

l.263: Hook -> Hooke **Corrected.**

l.291: XOZ region Please clarify! Do you mean an XZ cross section at y=0? **XOZ is changed to "atmospheric".**

l.304: produces -> produced **Corrected.**

l.310: Hook -> Hooke **Corrected.**

l.420: paper number or page range missing in the reference Dalin et al. (2016) **The page range is added.**

l.423: paper number or page range missing in the reference De Angelis et al. **The paper number is added.**

l.426: reference Djuth et al. (2004): journal should be GRL, not JGR. **The journal title is corrected.**

l.448: reference Godin: Earth Planets Space, 67, 47, ...?? **47 is the paper number.**

l.466: L"uhr, -> L\"uhr, **Corrected**

l.460: Yigit, **Corrected**

l.475: reference Rapoport: please delete ", Elsevier, 2004," **Corrected**

l.487: last excess -> last access **Corrected**

l.493: volume number and pages are missing in the reference Yigit and Medvedev **Corrected**

l.494: reference Yigit and Medvedev is from 2015, not from 2014 **Corrected**

**Yours sincerely. Nikolai M. Gavrilov, Sergey P. Kshevetskii, and Andrey V. Koval**